# Evolution of A bHLH Interaction Motif

**DOI:** 10.3390/ijms22010447

**Published:** 2021-01-05

**Authors:** Peter S. Millard, Birthe B. Kragelund, Meike Burow

**Affiliations:** 1The Linderstrøm-Lang Centre for Protein Science, Department of Biology, University of Copenhagen, DK-2200 Copenhagen N, Denmark; peter.millard@bio.ku.dk (P.S.M.); bbk@bio.ku.dk (B.B.K.); 2REPIN and Structural Biology and NMR Laboratory, Department of Biology, University of Copenhagen, DK-2200 Copenhagen N, Denmark; 3DynaMo Center, Department of Plant and Environmental Sciences, University of Copenhagen, 1871 Frederiksberg C, Denmark; 4Copenhagen Plant Science Centre, Department of Plant and Environmental Sciences, University of Copenhagen, 1871 Frederiksberg C, Denmark

**Keywords:** MYB, bHLH, MYC, transcription factor, short linear motif (SLiM), intrinsically disordered protein (IDP), protein interaction, interaction motif, plant defense metabolism, jasmonate signaling

## Abstract

Intrinsically disordered proteins and regions with their associated short linear motifs play key roles in transcriptional regulation. The disordered MYC-interaction motif (MIM) mediates interactions between MYC and MYB transcription factors in *Arabidopsis thaliana* that are critical for constitutive and induced glucosinolate (GLS) biosynthesis. GLSs comprise a class of plant defense compounds that evolved in the ancestor of the Brassicales order. We used a diverse set of search strategies to discover additional occurrences of the MIM in other proteins and in other organisms and evaluate the findings by means of structural predictions, interaction assays, and biophysical experiments. Our search revealed numerous MIM instances spread throughout the angiosperm lineage. Experiments verify that several of the newly discovered MIM-containing proteins interact with MYC TFs. Only hits found within the same transcription factor family and having similar characteristics could be validated, indicating that structural predictions and sequence similarity are good indicators of whether the presence of a MIM mediates interaction. The experimentally validated MIMs are found in organisms outside the Brassicales order, showing that MIM function is broader than regulating GLS biosynthesis.

## 1. Introduction

In plants, many transcription factors (TFs) function in a combinatorial manner, where TF interactions serve to integrate internal and external cues to modulate gene expression levels [1]. Interactions between proteins from the two most expanded plant TF families, the MYB (v-myb avian myeloblastosis viral oncogene homolog) family and the bHLH (basic helix-loop-helix) family, are common, and together they regulate a large variety of biological processes [2,3]. Recently, we determined the binding site responsible for interactions between eight *Arabidopsis thaliana* MYB TFs and their bHLH partners, MYC2, MYC3, and MYC4 [4]. The binding site constitutes a short linear motif (SLiM) and coincides with a sequence motif defining a subgroup of six MYB TFs (MYB28, MYB29, MYB76, MYB34, MYB51, and MYB122) that require MYC-interaction for activity [4,5]. All members of this subgroup are involved in regulating glucosinolate (GLS) biosynthesis [6,7], and have homologs throughout the Brassicaceae [8]. Two additional MYB TFs in *A. thaliana* contain the MYC-interaction motif (MIM), MYB47 and MYB95. Their biological functions are less studied, although MYB47 was recently implicated in the regulation of seed longevity [9]. The MIM consists of the six residues [L/F]LN[K/R][V/L]A, with the core motif (residue positions that are absolutely required) being xLNxxA. Mutations in these core residues abolish the interaction [4].

Plant MYB TFs contain a DNA-binding domain (DBD) composed of one (1R MYB), two (R2R3 MYB), three (3R MYB) or four (4R MYB) MYB repeats, each comprising ~50–55 residues that fold separately into a three-helix structure [10,11,12]. Most common are the R2R3 MYB TFs, termed like this as the repeats are most similar to the second and third repeats in vertebrate MYB TFs, which have three repeats [13]. Apart from their DBD, TFs (including R2R3 MYB TFs) generally contain extensive intrinsically disordered regions (IDRs) [14,15,16,17,18] that provide specific molecular functions modulating TF activity. For example, SLiMs within IDRs facilitate interactions with SLiM-binding pockets in globular domains [19]. The MIM is a SLiM located outside the DBD [4], and the short length combined with its presence in a long disordered context increases the chance of *ex nihilo* evolution [20], i.e., the likelihood that proteins from other families could evolve an analogous interaction motif.

The specialized metabolites GLSs evolved in the ancestor of Brassicales ~90 million years ago [21,22]. They accomplish several specific functions, including defense against microbial pathogens and animal herbivory [23,24,25], but also regulate root growth and development [26,27,28,29]. Although they are constitutive defense compounds, GLSs can be induced upon stimulation, especially through jasmonate (JA) signaling upon herbivore and pathogen attack [6,30,31]. MYC2, MYC3, and MYC4 regulate JA-responsive genes, as JA signaling relieves repression of MYC2, MYC3, and MYC4 [32,33,34]. The molecular function of the MIM is to mediate interaction with MYC TFs and thereby link activity to JA signaling.

JA signaling has an established role in regulating plant defense and immunity [35]. In contrast to the evolutionarily young GLSs, the molecular components required for JA biosynthesis, signaling, and perception, including homologs of MYC2, are much more widespread in the plant lineage, being omnipresent in land plants [36,37]. Although the eight known MIM-containing TFs from *A. thaliana* are all R2R3 MYB TFs, and the MYC-interaction is essential for activating GLS biosynthesis [4], the direct role of the MIM is to link TF activity to JA signaling through MYC-interaction. We therefore hypothesized that MIMs could be present in proteins from other families, and in TFs that are not involved in GLS regulation but instead regulate other JA-responsive genes.

Two central questions arise as to whether MIMs are present in non-GLS producing organisms, and whether MIMs are present in proteins outside the R2R3 MYB TF family. Answers to these questions will contribute to a far better understanding of how SLiM interactions in TFs evolve and how they develop to link activity to pre-existing signaling networks. Using bioinformatics approaches, we identified and validated MIM-containing R2R3 MYB TFs throughout the angiosperms, including in non-GLS-containing plants. We further identified putative MIMs in trihelix TFs, located in a protein region compositionally dissimilar from the MIM-containing region of R2R3 MYB TFs. The trihelix TFs did not interact with MYC4 in our assay, which we rationalize through sequence analysis combined with biophysical experiments.

Together, our results show that the MIM is found in non-GLS producing organisms spread throughout the angiosperms, implying that the MIM is involved in regulating other biological processes besides GLS biosynthesis.

## 2. Results

### 2.1. Discovery of New MIMs

Initially, we wanted to investigate whether MIM-containing proteins exist outside the Brassicales, as this would imply that the MIM is involved in other biological processes besides regulation of GLS biosynthesis. We applied different search strategies and databases and invested several types of sequence information as a query. One approach was to use the full-length sequences of the eight AtMYB TFs with confirmed MIMs, and PHI-BLAST patterns such as [LVFI]LN[KR][IFLV]A. Using different PHI-BLAST patterns allowed fine-tuning the tolerance for variation at the different positions in the motif. As the full-length protein sequence including the DBD was used as a query, this approach yielded significant hits only in other proteins from the same family (the R2R3 MYB TF family). Yet, as a binding site that provides a regulatory link to JA signaling, the MIM might be relevant in other protein families.

Therefore, to circumvent this limitation, an additional approach was used, where an alignment of the motif region only (from the eight MIM-containing AtMYB TFs) comprised the query (Figure 1a). This way we did not limit the search to protein types similar to those we already knew, but instead focused on the motif itself. The resulting query sequence is short and has some degeneracy at non-core positions (only three residues in the xLNxxA core motif were strictly required for interaction [4]). Further, not to disregard information from the flanking regions [38], we performed a number of those searches, where the alignment used as query included between 0 and 20 flanking residues on each side [39]. The use of iterative search tools, such as jackhmmer, that repeatedly adjusts the query according to significant hits, further facilitated the discovery of more diverged candidates.

We initially performed taxonomy-restricted searches, excluding all sequences from Brassicales, as the already known MIM-containing proteins from *A. thaliana* and their homologs from other GLS-producing plants otherwise dominated the search results and the readjustment of the hmm query in the iterative searches.

Outside angiosperms (flowering plants), we identified only a few significant hits. They were sporadic and all contained repetitive units with similarities to the motif. Further, these motif hits were present in unrelated proteins, in species where we could not find homologs of MYC TFs. Two examples of this are (i) (UniProt IDs) A0A0L7LIE3 from *Operophtera brumata* (winter moth), a putative α/β hydrolase, which had 16 hit-instances, and (ii) I4N4U3 from the bacterium *Pseudomonas sp. M47T1*, an uncharacterized protein with 6 hit-instances, showing 49% sequence identity to A0A1B4WVT0, a putative transferase. Most likely, these were significant hits because they contained a higher number of instances that aligned to the query but were disregarded because of the absence of MYC TFs in these species.

Within angiosperms (still excluding Brassicales), we found numerous significant hits to investigate further. Representative examples of our findings are shown in Figure 1 (see Table 1 for more information on each hit). Most of the putative MIM-containing proteins we identified were R2R3 MYB TFs. Within the Rosids, we could find MIM-containing R2R3 MYB TFs in most, but not all, orders (e.g., not found in Myrtales). We also found few R2R3 MYB TFs with a putative MIM in the Asterids, whereas other species with published annotated genomes in this clade (e.g., *Nicotiana benthamiana* [40]), gave no hits. The failure to identify putative MIM-containing proteins in certain organisms included in the databases does not necessarily imply their absence in that organism, as genomic information may be missing, genomes may be poorly annotated, or the instance simply fell below the significance threshold. However, our findings suggest that putative MIM-containing R2R3 MYB TFs are widespread in the eudicots.

Basal eudicots are separated from the core eudicots by more than 100 million years of evolution [42]. The Ranunculales order is the most basal clade in the basal eudicots, hence the sister group to the remaining eudicots [43]. Within this order, we found a single significant hit, in an R2R3 MYB TF of *M. cordata* (McR2R3MYB). Besides the R2R3 MYB TFs, we identified hits in proteins belonging to a few other protein families. These were all TFs, although we did not impose that limit by our search criteria.

First, in several species, we identified trihelix TFs with a putative MIM near their C-terminal. Trihelix TFs have low sequence similarity to MYB TFs, and are usually not included in the MYB family although their DBD bears structural resemblance [44]. While the helices are longer in trihelix TF DBDs, they form a similar triple helix structure that interacts with DNA in a sequence-specific manner [44]. As the searches resulting in putative MIM hits amongst trihelix TFs had been restricted to exclude Brassicales, we used these sequences as queries to search for homologs in Brassicales. This resulted in the identification of the *A. thaliana* trihelix TF ASR3 (UniProt ID Q8VZ20), with a putative MIM near the C-terminus, like the trihelix TFs identified in other organisms.

Second, in Myrtales, an order that is evolutionarily rather close to the Brassicales, we did not find putative MIM-containing R2R3 MYB TFs. However, we found a significant hit within the bHLH DBD of a bHLH TF. In this case, we were not able to find a putative MIM-containing homolog in *A. thaliana* by using the bHLH sequence to search back against Brassicales. Members of the bHLH family have a DBD structurally dissimilar to the MYB DBD, and bind DNA in a different manner [45]. As the confirmed MIMs are SLiMs [4] that are accessible because of their location within IDRs, we did not further consider this putative MIM as it was found within a conserved, highly structured domain.

Third, apart from trihelix TFs with putative MIMs, the only significant hit outside the eudicots that we considered further was a 3R MYB TF in the Poales order. Like the MIMs we found in R2R3 MYB TFs, the putative MIM in this 3R MYB TF is located outside the DBD. Similar to the bHLH hit, we were not able to find Brassicales homologs of this protein with a putative MIM.

Thus, although GLS biosynthesis is the main biological process known to be regulated by the characterized MIM-containing proteins found in *A. thaliana*, we were able to locate putative MIMs in proteins of many species that do not produce GLSs (Table 1). Further, our approach enabled us to find a putative MIM in an additional *A. thaliana* protein, belonging to a different protein family, namely the trihelix TF family.

Recently, tools dedicated to SLiM discovery have been developed, for instance, SLiMSearch [46] and PSSMSearch [47]. They improve SLiM discovery by including attributes such as sequence conservation of core positions versus the surroundings, structure propensity and accessibility, sequence context, ontology, and known interaction data, directly when filtering and ranking putative hits. They are so far limited to fewer target organisms, which for Viridiplantae (phylogenetic group that contains all land plants and green algae) means only *A. thaliana* and *Physcomitrella patens*. Our searches with these tools in *A. thaliana* revealed no new hits with SLiMSearch, but numerous putative hits using PSSMSearch (Appendix A). However, ASR3 was not detected, and neither were seven out of the eight R2R3 MYB TFs used as a query, which are known to contain the motif and to interact with MYC TFs. Of these, only MYB122 came above the disorder threshold. These tools are, however, currently not suitable for addressing the main objective of this study, namely whether MIM-containing proteins are present in non-GLS producing organisms, and thus involved in other biological functions.

For the following analysis, we focused on the putative MIM-containing proteins found outside Brassicales, and ASR3, as it has homologs with putative MIMs throughout the angiosperms.

### 2.2. Putative MIMs in Different Protein Families are Present in Dissimilar Sequence Contexts

The MIM is a SLiM [4], i.e., a short sequence constituting a binding motif within a larger IDR. Thus, we may expect the newly discovered MIMs to be located within similar disordered regions. Although the core positions contribute most of the affinity and specificity of SLiM-mediated interactions [19], the flanking regions and general sequence context might allow/disallow, or at least modulate, interaction [39]. A previous attempt to introduce a functional MIM by changing a few residues in the IDR of AtMYB75 [4] indicated that there are decisive attributes besides just the xLNxxA core motif. Therefore, we could further hypothesize other MIM-containing regions to have similar chemical properties. On the other hand, previous results have shown that non-core residues and certain general properties of the motif flanking region (such as high positive charge), although conserved amongst the proteins from *A. thaliana*, were not required for interaction [4]. We selected six putative MIM-containing proteins for further analysis (named in Figure 1, bold in Table 1), to simultaneously capture the different protein families and cover a broad evolutionary distance. To assess chain properties, the proteins were first analyzed using several disorder predictors, and to visualize local compositional biases to detect disordered domains, we used IDDomainSpotter [48]. This tool calculates the relative fraction of residues within a sliding window, some residues yielding a positive contribution (+) and some a negative contribution (−) and hence allows comparison of structural and chemical properties between proteins.

Three features distinguish the MIM region in the confirmed MIM proteins [4,48] as shown here for MYB29 (Figure 2). First, the MIM coincides with local order (seen as a dip in disorder profile) within a large IDR (the region N-terminal to the MIM has been experimentally verified to be highly disordered and dynamic [48]). Second, the MIM region has an overall positive net charge (+RK−DE) compared to surrounding regions. Third, the MIM is flanked by regions with a compositional bias towards certain small, hydrophilic, and disorder-promoting residues and few positively charged residues (+PST−RK). These features were shared amongst all MIM regions in the newly discovered R2R3 MYB TFs (Figure 2 and Appendix A).

For the 3R MYB TF from *Brachypodium distachyon*, the MIM was also located within a large IDR, although not associated with a local dip in disorder (Appendix A). For both 3R MYB and trihelix TFs, the MIM region was not characterized by the same compositional biases (as per the IDDomainSpotter profiles) as the MIM in R2R3 MYB TFs.

Two observations on the trihelix TFs draw special attention. First, the annotation of the trihelix domain by PROSITE does not cover the entire region suggested to be structured by the disorder predictors. Different domain databases are not in agreement for this class of trihelix TFs: for PROSITE [41] used in Figure 2, the DBD covers residues R38 to A104, while Pfam [54] indicates the residues L36 to N129 as belonging to the DBD. However, only a single three-dimensional structure of this class of DBDs has yet been determined [44], but its helices only cover parts of the alignment, leaving the conundrum unresolved. Second, the putative MIMs in the trihelix TFs were predicted to be within a more ordered region, although the different disorder predictors disagree, especially for the trihelix TFs from *Ananas comosus* and *Solanum pennellii* (Figure 2 and Appendix A). The confirmed MIMs were predicted to be located within IDRs, and it is therefore possible that, similar to most SLiMs [19], accessibility and flexibility is required for interaction, although this has yet to be confirmed by experiment.

To sum up, the discovered MIM-containing R2R3 MYB TFs share context characteristics with the confirmed MIM proteins, whereas the other protein families differ. Therefore, it is essential to validate our search by testing whether the newly identified putative MIMs can mediate interactions with MYC TFs.

### 2.3. Only MIMs in R2R3 MYB TFs Mediate MYC-Interaction

We next performed interaction assays with each of the R2R3 MYB and trihelix TFs shown in Figure 2 and Appendix A, against MYC4 from *A. thaliana* (Appendix A) or MYC3 and MYC4 (Figure 3, Appendix A). Interaction was tested using a chemically-inducible split-ubiquitin system [55], and as a negative control we used AtMYB75, which does not have a MIM. All of the R2R3 MYB TFs tested interacted with MYC4 (Figure 3 and Appendix A). This shows that MIM-containing R2R3 MYB TFs are present in organisms as diverse as *T. cacao*, *G. max* and *M. cordata*.

For physiological relevance, MIM-containing proteins must encounter interaction partners in vivo. As the molecular constituents of JA signaling, including MYC2 homologs, are conserved in land plants [36,37], the species where we identified putative MIMs should also contain MYC TFs. Therefore, to extend our analysis to a physiologically relevant MYC TF, we performed a BLAST search using AtMYC2, AtMYC3 or AtMYC4 protein sequence as query against *M. cordata* protein sequences. All three queries result in the same top candidate: (UniProt ID) A0A200PXR5 (>50% sequence identity to each AtMYC TF), here termed McMYC. We tested interaction between McMYC and McR2R3MYB and four different MYB TFs from *A. thaliana*. In this split-ubiquitin system MYC2 is auto-activating as a bait, enabling growth on selection media even when the prey plasmid is empty. McMYC, like AtMYC2, resulted in auto-activation (Appendix A), rendering us unable to draw any conclusions on its possible interactions.

The trihelix TFs we found, although containing the core motif residues, did not interact in our split ubiquitin assays (Figure 3, Appendix A), possibly because of different structural context. The growth on SD-AHLW + Rapa indicates that the fusion proteins are capable of interacting, i.e., that they are expressed, correctly folded, and located in the same subcellular compartment. Thus, our next question pertained to the structural propensities of the motif region.

### 2.4. Similar Pattern but Different Amplitude of Helical Propensity in MYB29-MIM and ASR3-MIM Peptides

As the trihelix TFs we discovered through our MIM search do not interact with MYC4 although the core positions are identical to those of the R2R3 MYB TFs, we asked whether a difference in secondary structure between the MIMs in MYB29 and ASR3 could explain this discrepancy. To answer this, synthetic peptides covering the MIM with flanking residues (16 residues in total) for MYB29 and ASR3 (Figure 4g) were subjected to structural analysis. Exploiting the natural abundance of ^13^C and ^15^N, we assigned the C^α^ chemical shifts of MYB29-MIM and ASR3-MIM from two-dimensional nuclear magnetic resonance (NMR) spectra (Figure 4a,c and Appendix A) and calculated the secondary chemical shifts (SCS) based on sequence-corrected random coil chemical shifts [56,57], reporting on structure propensities (Figure 4b,d). Both peptides formed transient helical structures (SCS C^α^ > 0.1), with a slight dip around residues L190-N191 for MYB29 and L300-N301 for ASR3 (the LN residues in the core motif xLNxxA). ASR3-MIM was substantially more helical (SCS C^α^ values reaching >0.75; ~25% helicity [58,59]) than MYB29-MIM (highest SCS Cα value at 0.36; ~12%). Further, for ASR3-MIM, helical propensity was observed throughout the entire length of the peptide, whereas for MYB29-MIM it was observed only for residues S185-A195.

Far-UV circular dichroism (CD) spectra revealed that both peptides exhibited a pronounced negative band at 200 nm originating from disordered structures, but also some negative ellipticity around 220 nm, indicative of residual helicity. The magnitude of these features supports a larger helix population in ASR3-MIM compared to MYB29-MIM.

Together, our results suggest that, although the core positions are conserved, and the peptides share a transient helical nature, the difference in the magnitude of their helicity and/or other dissimilar properties of their sequence context result in different interaction profiles. To further investigate the differences between this region in trihelix and R2R3 MYB TFs, we next turned to investigate their broader evolutionary conservation.

### 2.5. Conservation of MIM and Its Context in R2R3 MYB and Trihelix TFs

SLiMs are often distinguished by relatively high conservation within otherwise rapidly evolving protein regions [20]. Low conservation is considered typical of IDRs, although that is not always the case [60,61,62]. Observation of the evolutionary conservation of SLiMs and flanking regions thus sheds light on the probable importance of different sequence characteristics, such as e.g., compositional biases or structural propensities.

To this end, we collected the sequences of R2R3 MYB and trihelix TF homologs that were most similar to the MIM-containing proteins, originating from species spanning the different orders of the angiosperms, also shown in Figure 1, aligned them, and colored the alignment according to conservation (Figure 5).

The results reveal a pronounced difference between the motif regions of the trihelix and R2R3 MYB TFs. The flanking regions surrounding the motifs are much more conserved in the trihelix TFs than in the R2R3 MYB TFs. In fact, the putative motif in the trihelix TFs does not stand out as more conserved than the region it is contained within, as expected for a SLiM, and as it does for the R2R3 MYB TFs.

The difference in (i) secondary structure propensity of ASR3-MIM and MYB29-MIM peptides (Figure 4), (ii) sequence characteristics of the MIM-region (Figure 2), (iii) conservation of the MIM versus MIM region (Figure 5), and (iv) ability to interact with MYC3 and MYC4 (Figure 3), indicate that the trihelix TFs were false-positives in our search for MIM-containing proteins. These differences highlight both the importance of validating in silico findings and the power of sequence-based predictions and analyses of intrinsic disorder, compositional biases, and evolutionary conservation.

## 3. Discussion

### 3.1. SLiM-Hunting Challenges

SLiMs are defined as short (3–10 residue) segments in IDRs that mediate interactions with SLiM-binding pockets [19]. SLiM-binding pockets often have several or many interaction partners that constrain the shape and characteristics of the pocket, and are consequently conserved over large evolutionary distances [20]. That only three or four residues define the SLiM makes them challenging to identify [46], as sequence databases of a certain size would by chance contain segments that match and give rise to false positives. Statistical descriptors, such as the E value, help to assess the risk of such false positives and to filter out irrelevant results. However, if the assumptions leading to the query are correct, e.g., only very few positions and sequence characteristics are required for interaction, this will likewise lead to filtering out relevant hits as false negatives. Still, predictions alone cannot be used for SLiM identification.

In the present work, we addressed these challenges by performing many searches with different information in the query. Further, iterative searches concede, and to some degree, make up for the possibility that different sequence properties may have arisen along evolutionary paths leading to other MIM-containing proteins than those known in advance. Finally, we proceeded to validate our hits by experiment.

Experimental validation is especially important for hits that either differ in the properties expected to allow interaction, such as accessibility or sequence composition, or when the biological relevance is not obvious—for example, when the hit is present in a species lacking the expected binding partner candidate or where the inferred biological function is absent. In this work, we identified hits that could belong to both of those cases: trihelix TFs have their putative MIM within a different sequence context, and all of the new MIM-containing MYB TFs identified are present in species lacking GLSs.

The trihelix TFs we identified turned out to be false positives, highlighting the importance of local structure and sequence composition in the MIM region, and thus the power of sequence predictions when comparing proteins. When the sequence and structure predictions of putative MIMs matched those of confirmed MIMs, we were able to validate MYC interactions experimentally (Figure 3 and Appendix A). That the MIM is functional in several R2R3 MYB TFs outside Brassicales indicates that the biological function of the MIM is broader than regulation of GLS biosynthesis.

Most of the hits found using PSSMSearch (Appendix A)—like the trihelix TFs—have a different sequence composition in the motif flanking region (e.g., not an overrepresentation of small hydrophilic or positive residues), and residues with different properties than the experimentally validated MIMs at non-core positions, e.g., negatively charged, bulky, structure-disruptive (Pro), or aromatic. Although the core residues (those that by themselves disrupt interaction upon mutation), and a few other hypothesis-driven sequence characteristics (e.g., positive charges) have been investigated [4], we are still unaware of residues that are *disallowed* at certain positions. This information would facilitate MIM discovery especially when using tools such as PSSMSearch. Still, these hits were filtered by disorder (the default disorder cut-off that also filtered out ASR3 and seven out of eight known MIM-containing R2R3 MYB TFs in *A. thaliana*). Whether any of the hits found using PSSMSearch (Appendix A) can interact with MYC TFs is not known.

Although not indicated as a domain by either PROSITE and Pfam, the C-terminal predicted-to-be-ordered region in ASR3 and the other identified trihelix TFs, is conserved among trihelix TFs from the SH4 clade (which ASR3 belongs to) [63]. The structure has not been determined experimentally, but it has been proposed to form a coiled-coil responsible for homodimerization. ASR3 does homodimerize, a process that was blocked when residues K255 to Q288 were deleted [64] (the putative MIM covers _299_VLNKLA_304_). That the entire region is conserved (Figure 5) and not involved in MYC interaction (Figure 3) suggests that the presence of many instances matching the core motif was not because of selection for MYC interaction, but instead because of other evolutionary constraints, possibly of structural origin.

### 3.2. Biological Function of MIM-Containing R2R3 MYB TFs not Involved in GLS Regulation

We identified and validated several MIM-containing TFs from species that do not produce GLSs (Figure 1 and Table 1). Furthermore, two additional R2R3 MYB TFs from *A. thaliana*, MYB95 and MYB47, contain a MIM. The MIMs present in TFs of non-GLS producing plants (and MYB47 and MYB95) may be involved in regulating a more ancestral biological process. Since JA has an established role in defense [35], and the MIM offers a direct regulatory connection to JA signaling, this could, for example, be biosynthesis of other defense compounds. However, JA also plays roles in regulation of e.g., abiotic stress responses [65] and plant development, including reproductive organs [66]. MYB47 is involved in seed longevity [9]. Whether interactions between MYB47 and MYC TFs are involved in coordinating JA signaling into the regulation of seed longevity is unknown.

MYC2 has been suggested to be a hub that interacts with countless other proteins to fine-tune most JA responses [67]. None of these already known interaction partners were identified by our search for new MIM instances, suggesting that they use other binding sites to interact with MYC2.

### 3.3. Evolutionary History of the MIM

The short length of SLiMs increases their chances of evolving convergently, as they may arise from only a few base substitutions within an IDR. Convergent evolutionary events challenge the establishment of evolutionary relationships. We identified putative MIMs in R2R3 MYB TFs of several species spread throughout the eudicot lineage (Figure 1). Furthermore, through experiment, we validated that out of three tested newly discovered MIM-containing R2R3 MYB TFs, all interacted with AtMYC4 (Figure 3). Confirmed MIM-containing R2R3 MYB TFs thus belong to species as diverse as *A. thaliana*, *T. cacao*, *G. max*, and *M. cordata*. If all the R2R3 MYB TFs we found (Figure 1 and Table 1) share evolutionary origin, this would suggest that the MIM evolved before the split between basal and core eudicots, which is over 100 million years ago [42]. An important question is therefore why there are many species within the eudicots where we failed to identify MIM-containing proteins, even though we explicitly searched in several model organisms (for instance, *Populus trichocarpa*, *Nicotiana benthamiana*, and *Solanum lycopersicum*). A combination of the following factors could explain this apparent discrepancy. (1) For many species, high quality annotated genomes are unavailable. Thus, they may still be present but remain to be annotated. (2) Our analysis may not have been exhaustive due to a too restrictive SLiM definition. (3) The SLiM may in some species have diverged at core SLiM positions, making them silent to our searches. (4) Some species could have lost putative MIMs or MIM-containing proteins. (5) The numerous instances of MIM-containing proteins could reflect a high possibility of convergence.

Several proteins confirmed to interact with MYC TFs have more than one putative MIM. These include AtMYB34 (_148_SGSARL**LN**RV**A**SKYAV_163_ and _192_SPTSTL**LN**KM**A**ATSVL_207_) and McR2R3MYB (_222_SSSARL**LN**KI**A**TQAAT_237_ and _262_LSSAQL**LN**KM**A**TQPAA_277_). Further, the R2R3 MYB TFs identified in Rosales each contain three putative MIMs (Figure 1). Whether just one or more MIMs are functional remains to be experimentally evaluated, but if more are functional this could further support that MIMs may evolve convergently, and possibly in some cases appear in multiples to tune MYB-affinity [68,69].

If by chance a motif evolves ex nihilo, which enables a new, beneficial interaction, the interaction may initially have binding characteristics (such as affinity and specificity) that are far from optimal [19]. Subsequent mutations in the core motif and in surrounding residues then fine-tune the interaction. We most frequently identified MIMs in R2R3 MYB TFs, which could suggest that these TFs have properties that facilitate the evolution of a MIM, such as specific protein segments where it can readily evolve. Alternatively, these TFs regulate pathways where the connection to JA signaling is beneficial. Further work is needed to determine whether the many MIM instances in R2R3 MYB TFs are the result of convergent evolution or a shared evolutionary origin.

### 3.4. Evolution of Regulation in Novel Biological Processes

Since the MIM defines a subgroup of MYB TFs that are all involved in GLS regulation in *A. thaliana*, and they have homologs throughout Brassicaceae, it seems that once GLS biosynthesis evolved, early species in this taxon or their ancestor implemented the MIM to regulate GLS biosynthesis. As JA signaling is elevated by e.g., wounding and herbivore feeding [70], this allowed induction by relevant environmental stimuli to control the biosynthesis of this novel defense compound. JA signaling is much older than MIMs [36,37], and other MYB TFs bypass MYC-interaction through direct interactions with JAZ proteins [71]. Thus, when first evolved, the MIM was likely employed as a new way to connect to JA signaling. Further study of the evolution of protein interactions such as these will reveal how organisms employ already existing interaction motifs to regulate newly evolved biological processes.

Ex nihilo evolution of SLiMs allows adding molecular function to previously non-functional regions of proteins [20], providing a resource for evolutionary plasticity through new regulatory connections. Disorder can be a driver of innovation and divergence in protein function, and IDRs are capable of rapidly rewiring their interaction networks, compared to structured protein domains [72]. R2R3 MYB TFs have extensive IDRs [14], like TFs in general [15,16,17,18]. Thus, the ability of SLiMs to evolve convergently may have been key for the development and expansion of combinatorial control of gene expression as observed through cross-family TF interactions in plants [1]. Exuberated combinatorial control of gene expression is essential for plants to conduct coordinated physiological responses depending on complex environmental conditions.

## 4. Materials and Methods

### 4.1. Bioinformatics

Several tools were applied to search for MIM-containing proteins, including PHI-BLAST (https://blast.ncbi.nlm.nih.gov/Blast.cgi) and jackhmmer (https://www.ebi.ac.uk/Tools/hmmer/). The searches were performed against several different databases (non-redundant protein sequences (nr), Reference Proteomes, Ensembl, and Ensembl Genomes), to cover as many organisms as possible (the nr database includes all non-redundant CDS translations of GenBank along with all RefSeq, UniProtKB/Swiss-Prot, PDB and PRF proteins [73]).

Disorder prediction was performed using the following webservers: IUPred2 [49], DISOPRED3.1 [50], PONDR VSL2 [51,52], and ODiNPred [53]. IDDomainSpotter [48] was used for calculating local compositional biases.

### 4.2. Cloning

Coding sequences with SfiI overhangs were ordered as synthetic gene fragments from Twist Bioscience (San Francisco, CA, USA), cloned into the pFRB (prey) or pFKBP12 (bait) vector [55] by restriction digest and ligation, and verified by Sanger sequencing (Eurofins Genomics, Ebersberg, Germany).

### 4.3. Split-Ubiquitin Assays

The NMY51 *TOR1-1* Δ*fpr1* yeast strain [55] was transformed according to DUALhunter™ instructions (DualsystemsBiotech, Schlieren, Switzerland). Overnight cultures (25 mL) in 2xYPAD were diluted to OD_600_ = 0.2 in 500 mL, incubated at 30 °C for 4–6 h with shaking 200 rpm, and sedimented at 700 *g* for 5 min. The yeast cells were washed in 300 mL MQ water, then 100 mL 0.1 M LiOAc, resuspended in 15 mL 0.1 M LiOAc with 15% (*v*/*v*) glycerol, aliquoted and stored at −80 °C. Double transformation was performed by thawing the cells on ice (50 μL per transformation), mixing with transformation mix (240 µL 50% (*v*/*v*) PEG-3350, 35 µL 1 M LiOAc and 25 µL 2 mg/mL ssDNA per transformation) and 200 ng each of the bait and prey plasmids, and transferred to a 42 °C water bath for 45 min. The cells were then chilled on ice, sedimented at 3000× *g* for 1 min, resuspended in sterile water and plated on SC-LW plates.

Three individual colonies of each bait/prey combination were inoculated in liquid SD-LW media and grown overnight at 30 °C with shaking 200 rpm. The cultures were adjusted to OD_600_ = 0.05 and spotted on plates (SD-LW, SD-AHLW and SD-AHLW with 10 µg/mL Rapamycin) in 10-fold dilution series (OD_600_ = 0.05, OD_600_ = 0.005, OD_600_ = 0.0005, and OD_600_ = 0.00005). The plates were examined and pictures taken after 2 or 3 days of growth at 30 °C.

### 4.4. CD and NMR Spectroscopy

Amidated and acetylated peptides MYB29-MIM (SSTSKLLNKVAARASS) and ASR3-MIM (DSLVAVLNKLADAVAK) were synthesized by TAG Copenhagen A/S (Copenhagen, Denmark).

For far-UV CD spectroscopy, the peptides were dissolved in 10 mM sodium phosphate pH 7.0 and centrifuged at 15,000× *g* for 20 min. The peptide concentration was estimated from absorbance at 214 nm [74] and the peptides were diluted to 60 µM with 10 mM sodium phosphate pH 7.0, and pH corrected to 7.0. Far-UV CD spectra were recorded from 260 nm to 190 nm at 20 °C on a Jasco J-815 CD spectropolarimeter with 2 s D.I.T., 1 nm bandwidth, and 0.1 nm data pitch at 10 nm/min. 20 scans were accumulated. Buffer spectra were recorded with identical settings and subtracted before analysis.

For NMR spectroscopy, ASR3-MIM was dissolved in 50 mM sodium phosphate pH 7.0. MYB29-MIM was dissolved in a small amount of 25% (*v*/*v*) acetic acid, before being diluted with 50 mM sodium phosphate, pH 7.0. The 1D ^1^H NMR spectrum of this sample was identical to that of a similar sample of lower concentration MYB29-MIM that failed to dissolve completely in 50 mM sodium phosphate pH 7.0. All NMR samples were recorded in 600 µL total volume, supplemented with 5% (*v*/*v*) D_2_O, 125 μM 2,2-Dimethyl-2-silapentane-5-sulfonate (DSS) and 0.02% (*w*/*v*) NaN_3_, and adjusted to pH 7.0. Final peptide concentrations were approximately 1.8 mM for MYB29-MIM and 0.9 mM for ASR3-MIM, as ASR3-MIM self-interacted at high concentrations. ^1^H-, ^1^H-^1^H TOCSY-, ^1^H-^1^H ROESY-, ^13^C-^1^H HSQC- and ^15^N-^1^H HSQC- spectra were recorded at 5 °C on a Bruker 800 MHz spectrometer. For MYB29-MIM, the ^1^H-^1^H TOCSY- and ^13^C-^1^H HSQC- spectra were also recorded at 25 °C to reveal the overlap between the lysine and arginine residues. Topspin (Bruker, Billerica, MA, USA) was used for recording, transforming and referencing the spectra. ^1^H chemical shifts were referenced directly to DSS at 0.00 ppm, and heteronuclei referenced indirectly by relative gyromagnetic ratios. CcpNmr analysis software [75] was used for manually assigning the spectra. Sequence-corrected random coil C^α^ shifts were calculated with correction for temperature and pH [56,57].

## Figures and Tables

**Figure 1 ijms-22-00447-f001:**
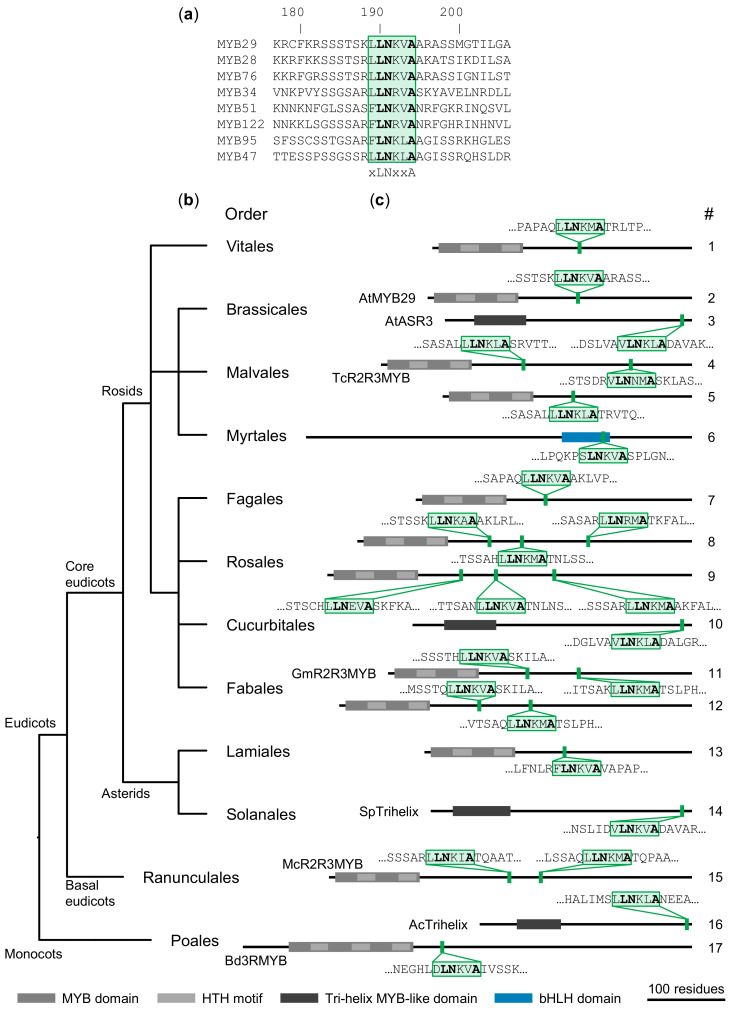
Examples of putative MYC-interaction motifs (MIMs) identified throughout the angiosperms. (**a**) Alignment of MIM-containing MYB transcription factors (TFs) from *A. thaliana*, with numbering according to AtMYB29, and with the MIM highlighted by a green box. (**b**) Phylogenetic tree showing the evolutionary relationship of orders with putative MIM-containing proteins. The tree was built with the NCBI Taxonomy Common Tree tool (https://www.ncbi.nlm.nih.gov/taxonomy) and is not to scale. (**c**) Domain organization and localization of each individual protein with a putative MIM. Domain information was obtained from PROSITE [41]. The location of the putative motif is shown in green, and the core motif residues (xLNxxA) in bold. Numbers (#) refer to each putative MIM-containing protein in Table 1.

**Figure 2 ijms-22-00447-f002:**
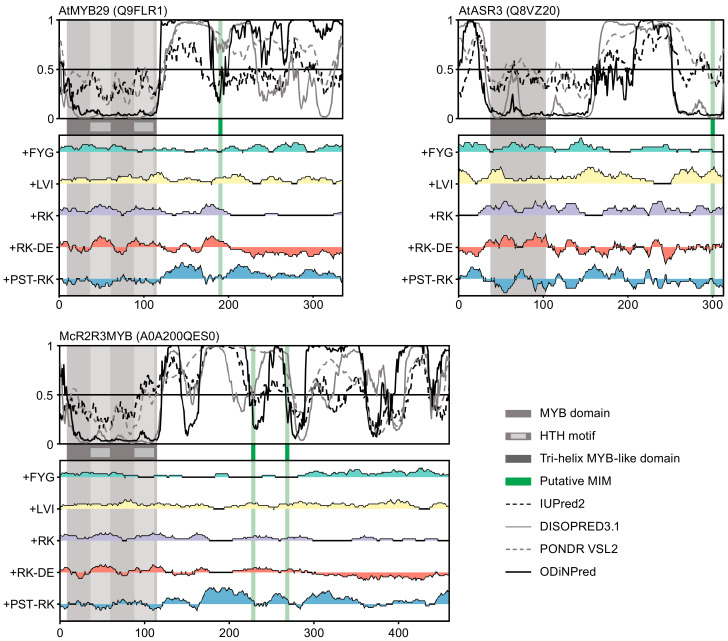
Disorder prediction and IDDomainSpotter profiles for putative MIM-containing proteins. Top panels show sequence-specific disorder predictions using IUPred2 [49], DISOPRED3.1 [50], PONDR VSL2 [51,52], and ODiNPred [53]. Bottom panels show IDDomainSpotter profiles: Each curve shows the fraction of a different set of residues within sliding windows of 15 residues [48]. Domain information was obtained from PROSITE [41]. The putative motif is shown by a green bar.

**Figure 3 ijms-22-00447-f003:**
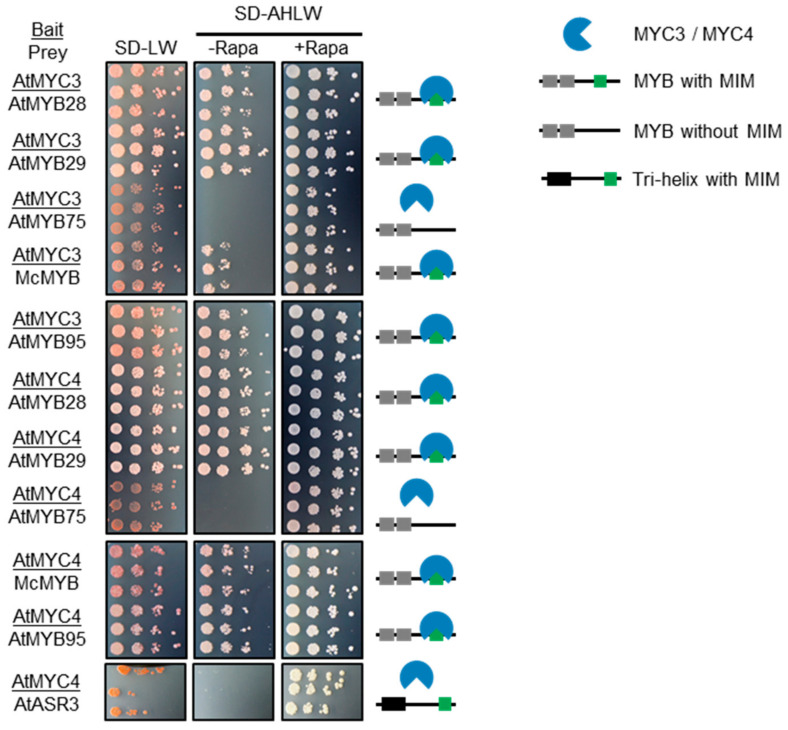
Testing interaction between *A. thaliana* (At) MYC3 and MYC4 against AtMYB28, AtMYB29, AtMYB75, AtMYB95, the MIM-containing trihelix MYB-like TF from *A. thaliana* (AtASR3), and MIM-containing R2R3 MYB TF from *Macleaya cordata* (McMYB). Growth on SD-LW medium confirms successful co-transformation with bait and prey construct and growth on SD-AHLW + Rapa medium indicates that bait and prey proteins are expressed and present in the same cellular compartment. The SD-AHLW-Rapa medium is selective for physical interaction of bait and prey protein. Rapa, rapamycin.

**Figure 4 ijms-22-00447-f004:**
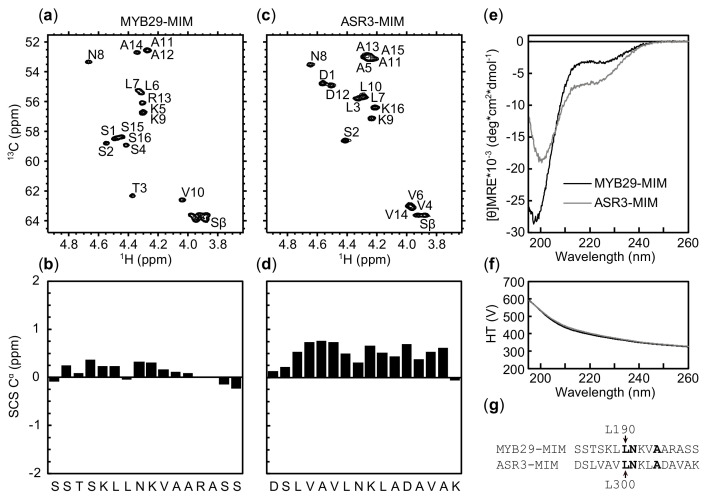
Structure elucidation of MYB29-MIM and ASR3-MIM. ^13^C-HSQC spectra of (**a**) MYB29-MIM and (**c**) ASR3-MIM with assignments. SCS of (**b**) MYB29-MIM and (**d**) ASR3-MIM. (**e**) Far-UV CD spectra and (**f**) HT voltage of MYB29-MIM and ASR3-MIM. (**g**) Sequence alignment of MYB29-MIM and ASR3-MIM (core motif residues in bold).

**Figure 5 ijms-22-00447-f005:**
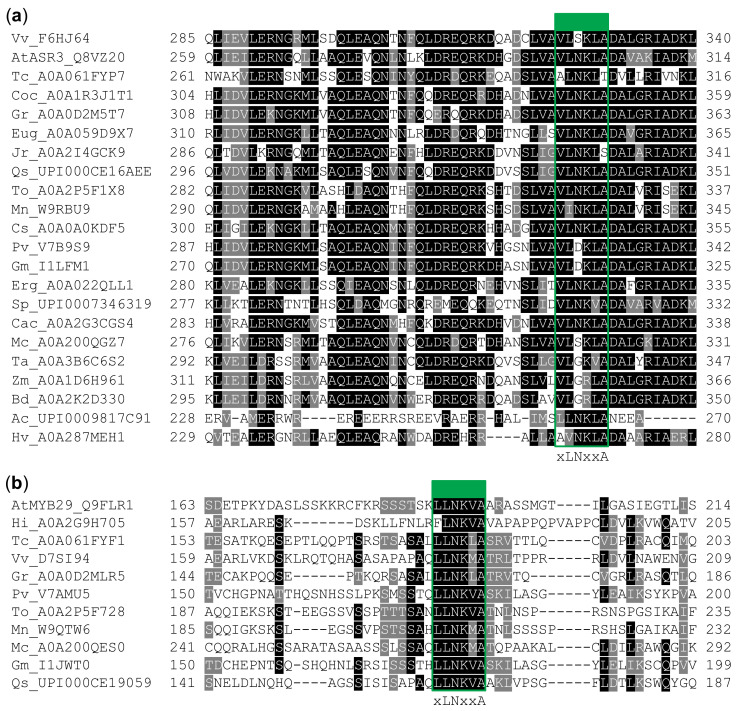
Alignments of (**a**) trihelix TFs and (**b**) R2R3 MYB TFs colored according to conservation, performed using BoxShade (https://embnet.vital-it.ch/software/BOX_form.html). For each position where a consensus (>50%) could be defined, residues identical to this consensus were colored black, while those similar were colored gray. A green border and box highlight the putative motifs. Vv: *Vitis vinifera*, At: *A. thaliana*, Tc: *Theobroma cacao*, Coc: *Corchorus capsularis*, Gr: *Gossypium raimondii*, Eug: *Eucalyptus grandis*, Jr: *Juglans regia*, Qs: *Quercus suber*, To: *Trema orientale*, Mn: *Morus notabilis*, Cs: *Cucumis sativus*, Pv: *Phaseolus vulgaris*, Gm: *Glycine max*, Erg: *Erythranthe guttata*, Sp: *Solanum pennellii*, Cac: *Capsicum chinense*, Mc: *Macleaya cordata*, Ta: *Triticum aestivum*, Zm: *Zea mays*, Bd: *Brachypodium distachyon*, Ac: *Ananas comosus*, Hv: *Hordeum vulgare*, Hi: *Handroanthus impetiginosus*.

**Table 1 ijms-22-00447-t001:** Putative MIM-containing proteins outside the Brassicales, and AtMYB29 and AtASR3 (from Figure 1). Bold: selected for further analysis.

#	Taxonomic Order	Species	Protein Type	UniProt/UniParc ID
1	Vitales	*Vitis vinifera*	R2R3 MYB	D7SI94
2 ^1^	Brassicales	*Arabidopsis thaliana*	R2R3 MYB	Q9FLR1
**3 ^1^**	**Brassicales**	***Arabidopsis thaliana***	**Tri-helix MYB-like**	**Q8VZ20**
**4**	**Malvales**	***Theobroma cacao***	**R2R3 MYB**	**A0A061FYF1**
5	Malvales	*Gossypium raimondii*	R2R3 MYB	A0A0D2MLR5
6	Myrtales	*Eucalyptus grandis*	bHLH	A0A059C313
7	Fagales	*Quercus suber*	R2R3 MYB	UPI000CE19059 ^1^
8	Rosales	*Morus notabilis*	R2R3 MYB	W9QTW6
9	Rosales	*Trema orientalis*	R2R3 MYB	A0A2P5F728
10	Cucurbitales	*Cucumis sativus*	Tri-helix MYB-like	A0A0A0KDF5
**11**	**Fabales**	***Glycine max***	**R2R3 MYB**	**I1JWT0**
12	Fabales	*Phaseolus vulgaris*	R2R3 MYB	V7AMU5
13	Lamiales	*Handroanthus impetiginosus*	R2R3 MYB	A0A2G9H705
**14**	**Solanales**	***Solanum pennellii***	**Tri-helix MYB-like**	**UPI0007346319 ^2^**
**15**	**Ranunculales**	***Macleaya cordata***	**R2R3 MYB**	**A0A200QES0**
**16**	**Poales**	***Ananas comosus***	**Tri-helix MYB-like**	**UPI0009817C91 ^2^**
17	Poales	*Brachypodium distachyon*	3R MYB	I1HT97

^1^ Proteins from GLS-producing plant species; ^2^ UniParc ID.

## Data Availability

The data presented in this study is contained within the article or Appendix A.

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
