# Peer review of "Evolution of A bHLH Interaction Motif"

_ijms, 2021, doi:10.3390/ijms22010447_

Round 1

Reviewer 1 Report

The study of Millard and colleagues is interesting and is really well done and well presented. In particular, they observed and demonstrated that MYC-interaction motif (MIM), typical of TFs usually related to regulation of GLS biosynthesis, is also found in non-GLS producing species, by a combinated approach of bioinformatics tools and experimental interaction assays. In my opinion the analytical pipeline is accurate and the adopted experimental procedure is coherent with the claims of authors.

This study is likely of great interest for the general plant research community focusing on MYB and bHLH TF families, very common and involved in a large variety of biological processes in the most of plant species.

For this reason, I think that the manuscript can be accepted for publication in IJMS

Author Response

We would like to thank the reviewer for going through our manuscript and for the very positive comments.

Reviewer 2 Report

This work first searched for proteins harboring the MYC-interacting motif (MIM) among broad range of organisms using a variety of informatic approaches, and reported that MIMs are found in some R2R3 MYB TFs of angiosperm species. Besides, it showed that MIM of the trihelix TFs could not interact with MYC and concluded that they are false positive by a combination of informatics and yeast experiment. This manuscript is logically constructed and the finding are important for understanding MYC-MYB interaction-mediated biological events, although authors might not obtain expected results. Here, I raised some points to be addressed before publication as below.

Title. I recommend next; “Evolution of a bHLH interaction motif in angiosperm”.

Table 1. It is more kind to distinguish GLS-producing species from non-producing ones. Please make it clear which species can produce GLS.

Line 241. “against MYC3 and MYC4” may be correct.

Line 265. “Figure 3” is wrong. “Supplemental Figure 2” may be correct. Supplemental Figure 2 is important for the downstream story. So, it is better to move it to a main figure or combine it with Figure 3.

Figure 3. It might be better to check if proteins with disrupted MIM (as a negative control) interact with MYCs. 

Author Response

We would like to thank the reviewer for the overall positive comments and for pointing out how to improve our manuscript. Here our responses to the specific points that were mentioned:

Comment: Title. I recommend next; “Evolution of a bHLH interaction motif in angiosperm”.

Response: It is striking that we found the MYC-interacting motif only in angiosperm species, which may , however, just reflect the low abundance of sequence information from Gymnosperms. As we consider our findings on the evolution of this SLiM to be general and thus of interest not only to the plant field, we would prefer to keep the original title.

Comment: Table 1. It is more kind to distinguish GLS-producing species from non-producing ones. Please make it clear which species can produce GLS.

Response: We agree with the reviewer and have added this information to Table1.

Comment: Line 241. “against MYC3 and MYC4” may be correct.

Response: We tested only one of the candidates against MYC3 and MYC4 and have clarified the statement accordingly.

Comment: Line 265. “Figure 3” is wrong. “Supplemental Figure 2” may be correct. Supplemental Figure 2 is important for the downstream story. So, it is better to move it to a main figure or combine it with Figure 3.

Response: The sentence indeed refers to Supplemental Figure 2 and we would like to thank the reviewer for pointing this out. We further agree that the data on trihelix-bHLH interaction are important for the story and have moved the assay showing the lack of interaction between AtMYC4 and the A. thaliana tri-helix ASR3 from Suppl. Figure 2 to Figure 3. We suggest to keep the data for the additional R2R3 MYB and tri-helix MYB-like transcription factors as supplemental information.

Comment: Figure 3. It might be better to check if proteins with disrupted MIM (as a negative control) interact with MYCs. 

Response: We had tested this important point previously and therefore did not include this control here. We have added this information in to the introduction (see l. 45).